# Molecular detection of *Bartonella* spp. DNA in dogs with hemangiosarcoma

**Cynthia Robveille**[1], **Ricardo G. Maggi**[1], **Erin Lashnits**[2], **Taryn A. Donovan**[3], **Keith E. Linder**[4], **Daniel P. Regan**[5], **Kevin D. Woolard**[6], **Edward B. Breitschwerdt**[1]*

**1** Department of Clinical Sciences, Intracellular Pathogens Research Laboratory, North Carolina State University - College of Veterinary Medicine, Raleigh, North Carolina, United States of America, **2** Department of Medical Sciences, University of Wisconsin-Madison - School of Veterinary Medicine, Madison, Wisconsin, United States of America, **3** Department of Anatomic Pathology, The Schwarzman Animal Medical Center, New York, New York, United States of America, **4** Department of Population Health and Pathobiology, North Carolina State University - College of Veterinary Medicine, Raleigh, North Carolina, United States of America, **5** Department of Microbiology, Immunology and Pathology, Colorado State University - College of Veterinary Medicine and Biomedical Sciences, Fort Collins, Colorado, United States of America, **6** Departments of Pathology, Microbiology and Immunology, University of California-Davis - School of Veterinary Medicine, Davis, California, United States of America

\* ebbreits@ncsu.edu

## Abstract

### Introduction

The potential role of pathogens, particularly vector-transmitted infectious agents, as a cofactor or cause of neoplasia has not been intensively investigated. We previously reported a potential link between *Bartonella* spp. bacteremia and splenic hemangiosarcoma (HSA) in dogs living in the United States. The purpose of this study was to: 1/ further determine the prevalence of *Bartonella* spp. DNA in dogs with splenic HSA from throughout the United States; 2/ assess the impact of sample preservation methods on *Bartonella* spp. DNA amplification using characterized tissue samples from dogs diagnosed with HSA.

### Methods

In a blinded manner, we determined the presence of *Bartonella* spp. DNA in scrolls from biorepository formalin-fixed paraffin-embedded (FFPE) spleens from dogs living in three distant locations geographically transecting the United States. DNA extracted from non-lesional spleens ($n = 249$), nodular lymphoid hyperplasia spleens ($n = 248$), and splenic HSA ($n = 330$) were tested by quantitative polymerase chain reaction (qPCR), and droplet digital PCR (ddPCR). Subsequently, *Bartonella* PCR results from FFPE tissues and formalin-fixed tissues were compared using previously tested fresh frozen tissues from an additional 48 dogs with HSA.

### Results

There was no significant difference in the proportion of *Bartonella* PCR positive FFPE tissues from dogs diagnosed with an alesional spleen, nodular lymphoid hyperplasia,

**Data availability statement:** All relevant data are within the paper and its Supporting Information files.

**Funding:** This research was supported by the American Kennel Club Canine Health Foundation (CHF Grant No. 02519, EBB and MB, http://www.akcchf.org/) and the State of North Carolina. The funders had no role in study design, data collection and analysis, decision to publish, or preparation of the manuscript.

**Competing interests:** In conjunction with Dr. S. Sontakke and North Carolina State University, Edward B. Breitschwerdt holds US Patent 7,115,385 Media and Methods for Cultivation of Microorganisms, which was issued on October 3rd, 2006. He is a co-founder, shareholder and Chief Scientific Officer for Galaxy Diagnostics, a company that provides advanced diagnostic testing for the detection of Bartonella spp. infections. Ricardo G. Maggi is a co-founder and the Chief Technical Officer for Galaxy Diagnostics Inc. All other authors declare no potential conflicts of interest with respect to the research, authorship, and/or publication of this article.

and splenic HSA. Regardless of the histological diagnosis, the most common *Bartonella* species identified was *B. henselae* (32/38). *Bartonella* spp. DNA was detected in a significantly larger proportion of fresh frozen tissues compared to FFPE tissues, when tested by qPCR (22/48 versus 1/48; $p$ <0.0001) or ddPCR (19/48 versus 1/48; $p$ <0.0001). Using ddPCR, *Bartonella* DNA was more often amplified from formalin-fixed tissues compared to FFPE tissues (15/39 versus 1/39; $p$ <0.0001). The sensitivity of qPCR on FFPE samples and formalin-fixed samples, when comparing to fresh frozen samples as the reference standard, was 4.5% and 11.8%, respectively.

## Conclusion

Due to decreased DNA amplification efficiency, FFPE scrolls should not be used for the detection of *Bartonella* infection in spleen samples from dogs with HSA. PCR testing of fresh frozen tissues substantially improves the detection of *Bartonella* spp. infection. If fresh frozen tissues are not available, formalin-fixed tissues should be tested with digital PCR to enhance *Bartonella* DNA detection.

## Introduction

*Bartonella* spp. are facultative intracellular, potentially zoonotic, Gram-negative bacteria, that are most often transmitted by arthropod vectors or animal bites and scratches [1]. To date, more than 50 species have been described. In dogs, these pathogens have resulted in granulomatous inflammation [2], and non-neoplastic vasoproliferative diseases, such as peliosis hepatis [3] and cutaneous angiomatosis [4]. Angiogenetic factors, like vascular endothelial growth factor, play a role in the pathogenesis of these diseases [5]. In addition, *Bartonella* DNA has been amplified from canine cancers, specifically perivascular wall tumors [5] and hemangiosarcoma (HSA) [6]. The latter is a common neoplasm in dogs, likely originating from hematopoietic stem cells or endothelial progenitor cells [7]. It generally affects middle-aged and neutered dogs, with German Shepherd Dogs, Golden Retrievers and Labrador Retrievers being overrepresented [8]. Common primary sites include the spleen, right atrium/auricle, liver, and skin/subcutis [9]. The prognosis of visceral HSA is poor because of a high rate of early and widespread metastases *via* hematogenous spread and local seeding after tumor rupture [10].

In a previous study performed in North Carolina, *Bartonella* spp. DNA was amplified more commonly from dogs with HSA (13/50; 26%) compared to nodular lymphoid hyperplasia (NLH) (5/50; 10%) and histologically unremarkable spleen from specific-pathogen free dogs (0/8; 0%) [11]. In that study, *Bartonella* spp. DNA was detected using conventional and real-time quantitative PCR from approximately 25 mg of splenic tissue manually excised from formalin-fixed paraffin-embedded (FFPE) surgical biopsy samples. A more recent study showed 32% (24/74) of fresh frozen biopsy samples from splenic HSA in dogs were *Bartonella* spp. PCR positive [6]. The use of FFPE tissue blocks has become widely used in molecular studies, as it represents the most frequently available resource for retrospective studies requiring prior histologic examination. Nevertheless, formalin induces DNA fragmentation, cytosine deamination, and cross-linking, that may negatively affect molecular analyses [12].

The aim of this study was to assess the proportion of dogs with *Bartonella* spp. DNA in splenic HSA *versus* non-neoplastic splenic specimens from three institutional-based pathology repositories. Given the generated results, we also evaluated the impact of sample preservation methods on *Bartonella* spp. detection in dogs with HSA, by comparing fresh frozen tissue, formalin-fixed tissue, and FFPE tissue from each dog. In addition to quantitative polymerase

chain reaction (qPCR), we used droplet digital PCR (ddPCR). This technique is based on partitioning of a single sample into 20,000 water-in-oil droplets, then each droplet undergoes a PCR simultaneously. It allows for absolute quantitation of target DNA molecules using binomial Poisson statistics, without the requirement for a standard curve [13].

## Materials and methods

### Sample collection

Formalin-fixed paraffin-embedded canine splenic samples were retrieved from pathology archive storage facilities, using the database from three veterinary institutions across the United States: The Schwarzman Animal Medical Center in New York (samples processed between 2006 and 2018), Colorado State University (samples containing at least 70% of viable tissue and processed between 2009 and 2019), and University of California at Davis. Samples were composed of alesional spleen ($n = 249$), NLH ($n = 248$), and HSA ($n = 330$). A single block was available per dog, and an H&E section from each block was independently reviewed by a board-certified pathologist (KEL) to confirm the histologic diagnosis. Demographic information and travel history were not obtained.

Given the generated results, tissues from 48 dogs diagnosed with HSA banked by the biospecimen repository of the Canine Comparative Oncology and Genomics Consortium were selected based on previously published *Bartonella* molecular results using fresh tissues snap-frozen in liquid nitrogen [6,14]. We randomly chose 26 dogs positive by qPCR and/or ddPCR, and 22 dogs negative by qPCR and ddPCR. Matched formalin-fixed tissue (stored in ethanol) and FFPE tissue were tested by qPCR and ddPCR. Formalin-fixed tissues were no longer available for 9 dogs.

### DNA extraction

FFPE tissues were deparaffinized using xylene and ethanol. DNA was extracted from three 50-μm sections of FFPE tissues and approximately 25 mg of formalin-fixed tissues using a Qiagen DNeasy® Blood and Tissue kit (Qiagen, Valencia, CA) following the manufacturer's instructions. To prevent cross-contamination, a new sterile scalpel blade (for formalin-fixed tissues) or a new microtome blade (for FFPE tissues) was used for each sample. Empty paraffin blocks were included and analyzed as controls for potential DNA carryover, as well as negative extraction controls. DNA yield and quality was assessed by spectrophotometry (Nanodrop, Wilmington, DE). The DNA was stored at -20 °C until needed.

### Molecular analyses

Both qPCR and ddPCR was performed in a blinded manner using primers targeting the 16S-23S rRNA intergenic transcribed spacer (ITS) region of *Bartonella* genus in conjunction with a BsppITS500 FAM-Taq Man® probe (5′ FAM-GTTAGAGCGCGCGCTTGATAAG -IABkFQ 3′; IDT® DNA Technology, NC, USA). Oligonucleotides BsppITS325s (5′ CTTCAGATGATGATCCCAAGCCTTCTGGCG 3′) and BsppITS543as (5′ AATTGGT GGGCCTGGGAGGACTTG 3′) were used as forward and reverse primers, respectively, as previously described [15]. Negative and positive controls were prepared using 5 μl of DNA from previously characterized healthy or positive dogs (clinical cases), respectively.

For qPCR, amplification was performed in a 25-μl final volume reaction containing 12.5 μl of SsoAdvanced™ Universal Probes Supermix (Bio-Rad, Hercules, CA, USA), 0.2 μl of 100 μM of each forward primer, reverse primer, and TaqMan probe (IDT® DNA Technology, Coralville, IA), 6.9 μl of molecular grade water (Genesee Scientific, San Diego, CA, USA), and 5 μl of DNA from each sample tested. Quantitative PCR was performed in an

CFX96® (Bio-Rad, Hercules, CA) under the following conditions: a single hot-start cycle at 95 °C for 3 min followed by 45 cycles of denaturing at 94 °C for 10 s, annealing at 68 °C for 10 s, and extension at 72 °C for 10 s. Amplification was completed by an additional cycle at 72 °C for 30 s. Positive amplification was assessed by analysis of detectable fluorescence signal vs cycle threshold values. Sequencing of all positive samples was performed at GENEWIZ Inc. (Research Triangle Park, NC, USA). Bacterial species and strain were defined by comparing similarities with other sequences deposited in the GenBank database using BLAST version 2.0.

The 20 μL final ddPCR reaction consisted of 5 μL of ddPCR™ Multiplex Supermix (Bio-Rad, Hercules, CA, USA), 0.2 μL of 100 μM of each forward primer, reverse primer, and TaqMan probe (IDT® DNA Technology, Coralville, IA, USA), 8.1 μL of molecular–grade water, 5 μL of sample DNA, 0.3 μL of 300 mM Dithiothreitol (DTT), and 1 μL of HindIII DNA restriction enzyme. The ddPCR analysis was performed using a QX One Droplet Digital PCR (Bio-Rad, Hercules, CA, USA) system under the following amplification conditions: a single hot-start cycle at 95 °C for 10 min followed by 40 cycles of denaturing at 94 °C for 30 s and annealing at 62.9 °C for 1 min. A final extension at 98 °C was performed for 5 min. Fluorescent droplet detection and distribution readings were recorded in channel 1. Due to instrument design limitations, digital PCR droplets are not able to be sequenced.

For the second part of the study, digital PCR amplification of the dog housekeeping gene BRAF was performed as amplifiable DNA internal control using the oligonucleotides CaFeBRAF-15s (5′-TCAYGAAGACCTCACAGTAAAAATAGGT-3′) and CaFeBRAF-110as (5′-GATCCAGACAACTGTTCAAACTGATG-3′) as forward and reverse primers, respectively, and oligonucleotides CaFeBRAF-50 (5′-Cy5.5-GTCTAGCCACAGTGAAATCTC GATG-BHQ_3–3′) as fluorescent probe (IDT® DNA Technology, NC, USA) [16]. Partitions were recorded in the orange channel.

## Study size

With a sample size of 240 samples per group (alesional spleen, NLH, and HSA), assuming based on previous literature [11] that >25% of HSA samples contained *Bartonella* spp. DNA, this study had 80% power to detect a statistically significant difference (alpha = 0.05) if the proportion of each control group containing *Bartonella* DNA was <15%.

When comparing sample preservation methods, assuming >50% of [reference group] contained *Bartonella* spp. DNA, this study had 80% power to detect a statistically significant difference (alpha = 0.05) if the proportion of [comparison group] containing *Bartonella* DNA was <20%.

## Statistical methods

The proportion of FFPE spleen samples from each diagnosis group (alesional, NLH, HSA) that had *Bartonella* spp. DNA amplified by qPCR or ddPCR was calculated; those proportions were compared using a chi-squared test of independence. The proportions of samples from each preservation method (fresh frozen, formalin-fixed, and FFPE) that had *Bartonella* spp. DNA amplified by qPCR or ddPCR was calculated; those proportions were compared using a chi-squared test of independence (or Fisher's exact test for small sample sizes). To compare the proportions of samples that were *Bartonella* PCR positive across geographic locations, we used chi-squared tests for each diagnosis group (alesional spleen, NLH, HSA) for qPCR and ddPCR. A Bonferroni correction for multiple comparisons was used, so for these comparisons statistical significance was set at *p* < 0.00833. To determine agreement between tests on two different samples from the same dog, the kappa statistic was calculated [17]. Statistical

significance was set at $p \leq 0.05$. All statistical analysis was performed in R v. 4.3.2 (R Core Team 2023).

### Ethics statement

Ethical approval was not required as this is a retrospective study using canine tissues either submitted for diagnostic pathology services to the Schwarzman Animal Medical Center, Colorado State University, and University of California, or from the biospecimen repository of the Canine Comparative Oncology and Genomics Consortium.

## Results

The proportion of FFPE tissues *Bartonella* spp. qPCR positive did not differ significantly between non-lesional spleen (13/249; 5.2%), NLH (14/248; 5.6%), and splenic HSA (11/330; 3.3%) ($p = 0.359754$) (Table 1). A similar trend with slightly higher proportion of positive samples was obtained with ddPCR (10.4%, 9.7%, and 11.2%, respectively, Table 1) ($p = 0.836694$). For each microscopic diagnosis, there was a statistically significant difference in the proportion of samples *Bartonella* spp. PCR positive between geographical locations, based on the locations of the submitting veterinary institutions, with ddPCR ($p = 0.0004$ for alesional spleens, $p = 0.0005$ for NLH, and $p = 0.0009$ for HSA) but not qPCR ($p = 0.0606$ for alesional spleens, $p = 0.3879$ for NLH, and $p = 0.1020$ for HSA).

Regardless of the histological diagnosis, based upon DNA sequence analyses, the most common *Bartonella* species identified was *B. henselae* (32/38, including 1 co-infection with *B. quintana*-like in HSA). Other *Bartonella* species included: *B. quintana* in 1 alesional spleen and 1 NLH, *B. koehlerae* in 1 NLH, and *B. chomelii* in 1 HSA. *Bartonella* species could not be determined for 1 NLH and 1 HSA.

The molecular prevalence of *Bartonella* spp. was significantly higher in fresh frozen tissues compared to FFPE tissues, with both qPCR (22/48 versus 1/48; $p <0.0001$) and ddPCR (19/48 versus 1/48; $p <0.0001$), and in formalin-fixed tissues compared to FFPE tissues with ddPCR (15/39 versus 1/39; $p <0.0001$) (Table 2). There was qPCR amplification of *Bartonella* DNA in FFPE tissues spiked with *Bartonella henselae* (results not shown), supporting the absence of an inhibitor of DNA amplification. The sensitivity of qPCR on FFPE samples, when comparing

**Table 1. Molecular results (qPCR, ddPCR) for the 827 FFPE tissues tested for *Bartonella* spp DNA.**

| Diagnosis | Location | qPCR+ | ddPCR+ | qPCR+ and ddPCR+ | qPCR+ or ddPCR+ |
|---|---|---|---|---|---|
| normal spleen | CSU ($n = 90$) | 6 | 18 | 2 | 22 (24.4%) |
| | AMC ($n = 66$) | 6 | 6 | 1 | 11 (16.7%) |
| | UC Davis ($n = 93$) | 1 | 2 | 1 | 2 (2.1%) |
| | Total ($n = 249$) | **13 (5.2%)** | **26 (10.4%)** | **4 (1.6%)** | **35 (14.1%)** |
| nodular lymphoid hyperplasia | CSU ($n = 76$) | 5 | 15 | 3 | 17 (22.4) |
| | AMC ($n = 77$) | 6 | 1 | 1 | 6 (7.8%) |
| | UC Davis ($n = 95$) | 3 | 8 | 1 | 10 (10.5%) |
| | Total ($n = 248$) | **14 (5.6%)** | **24 (9.7%)** | **5 (2.0%)** | **33 (13.3%)** |
| hemangiosarcoma | CSU ($n = 111$) | 6 | 21 | 5 | 22 (19.8%) |
| | AMC ($n = 88$) | 4 | 11 | 1 | 14 (15.9%) |
| | UC Davis ($n = 131$) | 1 | 5 | 0 | 6 (4.6%) |
| | Total ($n = 330$) | **11 (3.3%)** | **37 (11.2%)** | **6 (1.8%)** | **42 (12.7%)** |

CSU, Colorado State University; AMS, Animal Medical Center; UC, University of California

**Table 2. Molecular results (qPCR, ddPCR) for the 48 dogs diagnosed with HSA and selected on the basis of prior documentation of *Bartonella* spp PCR positivity or negativity. Residual formalin-fixed tissues were not available for 9 dogs.**

| | Fresh frozen *vs* FFPE tissues (*n* = 48) | | Fresh frozen *vs* FFPE *vs* FF tissues (*n* = 39) | | |
| --- | --- | --- | --- | --- | --- |
| qPCR+ | 22 | 1 | 17 | 0 | 5 |
| ddPCR+ | 19 | 1 | 15 | 1 | 15 |
| qPCR+ and ddPCR+ | 16 | 0 | 12 | 0 | 1 |
| qPCR+ or ddPCR+ | 25 | 2 | 19 | 1 | 19 |

FF, Formalin-fixed

to fresh frozen samples as the reference standard, was 4.5%. For dPCR, there was a larger number of partitions of the dog housekeeping gene (BRAFp) in the frozen samples compared to formalin-fixed tissues and FFPE tissues (results not shown).

*Bartonella* spp. DNA was amplified by qPCR from a significantly higher proportion of fresh frozen tissues compared to formalin-fixed tissues (17/39 versus 5/39; $p = 0.0025$), but not when tested by ddPCR (15/39 for both; $p = 1$). The sensitivity of qPCR on formalin-fixed samples, when comparing to fresh frozen samples as the reference standard, was 11.8%.

For qPCR results, there was slight agreement between results on samples from the same dog when comparing FFPE and fresh frozen tissue samples (kappa = 0.05, 95% CI -0.04–0.14) or formalin-fixed and fresh frozen tissue samples (kappa = -0.02, 95% CI -0.25–0.21). The proportion of samples that did agree (0.56 and 0.538, respectively) was not significantly higher than would be expected by chance alone (0.54, $p = 0.2719$; 0.548, $p = 0.8624$). For ddPCR results, there was slight agreement between results on samples from the same dog when comparing FFPE and fresh frozen tissue samples (kappa = -0.04, 95% CI -0.14–0.06). The proportion of samples that did agree (0.58) was not significantly higher than would be expected by chance alone (0.60, $p = 0.4134$). There was fair agreement between results on samples from the same dog when comparing formalin-fixed and fresh frozen tissue samples (kappa = 0.35, 95% CI 0.036–0.664). The proportion of samples that did agree (0.692) was significantly higher than would be expected by chance alone (0.527, $p = 0.0288$).

## Discussion

Antemortem diagnosis of bartonellosis remains challenging due to the bacteria's fastidious and slow-growing nature, low and intermittent bacteremia, and the lack of sensitivity of serological techniques currently available. These limitations have proven to be substantial when assessing *Bartonella* spp. infection in dogs with histological confirmation of HSA [6,14]. Also, due to current limitations with imaging modalities of *Bartonella* spp. with In Situ Hybridization and immunofluorescence, diagnosis of bartonellosis in surgical biopsy or postmortem tissue most often requires molecular testing [18–20]. Currently, despite known detrimental effects on PCR amplification sensitivity, many clinical, surgical and diagnostic workflows place ante- or post-mortem tissues solely into formalin for histologic evaluation, without consideration of the adverse effect in detection of microbial pathogens. As documented in this study, qPCR and ddPCR from FFPE tissues lowers *Bartonella* PCR sensitivity.

In this study, the proportion of samples with *Bartonella* positive qPCR from dogs diagnosed with splenic HSA or NLH from the three institutions was unexpectedly much lower compared to a previous study using FFPE tissues involving dogs with HSA from North Carolina [11]. As *Bartonella* infection in dogs appears to be ubiquitous across the United States [6], in retrospect, this discrepancy was likely related to differences in methodology

between studies. While FFPE tissues were used in both studies, paraffin scrolls (150 μm thick) instead of histologically selected core punches (25 mg) were used in this study, which resulted in a decreased tissue quantity per sample extraction. Thus, the quantity of extracted DNA (host and pathogen) potentially resulted in an underestimation of the actual proportion of *Bartonella* FFPE positive tissues. Because of DNA fragmentation in FFPE tissues, the use of amplification products smaller than 120–150 bp is recommended to maximize the number of templates to be used for PCR [21, 22]. Targeting *Bartonella* ITS gene sequences, we identified *Bartonella henselae* as the most common *Bartonella* species, as previously reported, including one study using fresh frozen samples [6,11]. As the generated amplicon for this species is around 140 bp, a minimal effect of DNA fragmentation for its detection is expected. Considering the rare flea infestations in Colorado, finding the highest proportion of samples *Bartonella* spp. ddPCR positive in the Colorado State University was surprising. Their samples had a short duration of fixation (≤ 24 hours) that could have positively impacted microbial detection.

The unexpectedly low amplification of *Bartonella* DNA from FFPE samples motivated our investigation of the impact of storage sample conditions in *Bartonella* spp. detection, using a previously characterized sample set [6]. Test results for matched FFPE and fresh tissues from 48 dogs, with a histological diagnosis of HSA, were compared for qPCR and ddPCR sensitivity. Among the 22 *Bartonella* qPCR+ dogs using fresh frozen tissues, only 1 dog remained qPCR+ using scrolls of FFPE tissue. Consequently, PCR using DNA extracted from paraffin scrolls does not appear suitable for *Bartonella* molecular testing. The sensitivity limitation of scrolls has been described for the detection of another intracellular pathogen, *Mycobacterium tuberculosis* [23]. Using this dog sample set, the molecular prevalence of *Bartonella* spp. was significantly lower in formalin-fixed tissues compared to fresh frozen tissues with qPCR, but did not significantly differ with ddPCR. This result could be explained by the higher sensitivity, especially in case of low bacterial load [14,24,25] and higher inhibitor tolerance of the ddPCR compared to qPCR [26]. Indeed, the ratio between target DNA to PCR reagents is substantially higher in ddPCR compared to qPCR. ddPCR has already shown its utility in the accurate quantitation of bacteria [27, 28]. It is worth noting that the template DNA volume, as well as the primers and probe sequences and concentrations, were identical for qPCR and ddPCR to avoid biases to the extent possible when comparing quantification of both assays. Obviously, by necessity, a different aliquot of extracted DNA had to be used for each comparative PCR reaction.

Interestingly, 6 of 15 dogs that were ddPCR+ in formalin-fixed tissues were previously ddPCR- in fresh frozen tissues. This finding was likely related to the low quantity and dispersed distribution of *Bartonella* within tissues, as well as testing only a single sample for each storage condition per animal. Using fresh frozen tissues, Lashnits et al. documented that *Bartonella* spp. DNA was ddPCR amplified from one of the two biopsy samples tested in 76% of bacteremic dogs with HSA, suggesting an increased diagnostic sensitivity of molecular testing when the latter is performed on multiple samples [14]. Similarly, sequential testing of human blood samples (obtaining 3 vs 1 sample) also increased the molecular detection of *Bartonella* spp. [29]. When attempting to confirm infection with a *Bartonella* sp., taking steps to optimize sensitivity of DNA detection should include testing multiple tissue locations and using duplicate from each tissue to accurately assess the prevalence.

FFPE DNA damage is influenced by many factors during processing and storage. Limitations in this study include unknown fixation time and duration between resection and formalin fixation, the variation in age of the paraffin blocks and storage parameters, all of which could have influenced the DNA quality and molecular results [30]. Different freezing methods were not assessed, as only samples snap-frozen in liquid nitrogen (subsequently

stored at -80° C) were available. Also, despite testing matched samples, the quantity and distribution of *Bartonella* likely varied among tissues selected for DNA extraction, as reported previously in cats with endomyocarditis-left ventricular endocardial fibrosis and *Bartonella* spp. infections [31].

In conclusion, FFPE scrolls should not be used for the detection of *Bartonella* infection in spleen samples from dogs with HSA. PCR testing of fresh frozen tissues substantially improves the detection of *Bartonella* spp. infection. If fresh frozen tissues are not available, testing core formalin-fixed tissues with droplet digital PCR should improve sensitivity over testing FFPE tissues.

## Acknowledgments

We thank Dr. Toni Richardson and Chance Liedig for their assistance in generating the molecular results in the Intracellular Pathogens Research Laboratory, North Carolina State University.

## Author contributions

**Conceptualization:** Edward Bealmear Breitschwerdt.

**Formal analysis:** Cynthia Robveille, Erin Lashnits.

**Funding acquisition:** Edward Bealmear Breitschwerdt.

**Investigation:** Cynthia Robveille, Ricardo G. Maggi, Keith E Linder.

**Methodology:** Ricardo G. Maggi.

**Resources:** Taryn A Donovan, Daniel P. Regan, Kevin D Woolard.

**Supervision:** Edward Bealmear Breitschwerdt.

**Writing – original draft:** Cynthia Robveille.

**Writing – review & editing:** Cynthia Robveille, Ricardo G. Maggi, Erin Lashnits, Taryn A Donovan, Daniel P. Regan, Edward Bealmear Breitschwerdt.

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
