## [Decision Letter · Decision Letter 0]

12 Mar 2025

PCR testing of fresh frozen tissues substantially improves molecular detection of Bartonella spp. DNA in dogs with hemangiosarcoma.

PONE-D-25-00356

Dear Dr. Breitschwerdt,

We’re pleased to inform you that your manuscript has been judged scientifically suitable for publication and will be formally accepted for publication once it meets all outstanding technical requirements.

Kind regards,

Dina Aboelsoued, Ph.D.

Academic Editor

PLOS ONE

Additional Editor Comments (optional):

Thank you for the submission of this comprehensive and well-written manuscript.

Reviewers' comments:

Reviewer's Responses to Questions

**Comments to the Author**

1. Is the manuscript technically sound, and do the data support the conclusions?

Reviewer #1: Yes

Reviewer #2: Yes

2. Has the statistical analysis been performed appropriately and rigorously?

Reviewer #1: Yes

Reviewer #2: Yes

3. Have the authors made all data underlying the findings in their manuscript fully available?

Reviewer #1: Yes

Reviewer #2: Yes

4. Is the manuscript presented in an intelligible fashion and written in standard English?

Reviewer #1: Yes

Reviewer #2: Yes

5. Review Comments to the Author

Reviewer #1: 1- The tittle need to change and the suggested tittle is:

' Molecular detection of Bartonella spp. DNA in dogs with hemangiosarcoma '

2- The abstract is clear.

3- The introduction is clear.

4- Methodology is enough to extract valuable data.

5- The results are clear

6- Discussion: is enough to extract valuable data

7- The authors in references part follow the style of the journal (Plos One). >>>>>>>>>>>>>>>>>>>>>>>>>>>>>>>>>>>>>>>>>>>>>>>>>>>>>>>>>>>>>>>>>>>>>>>>>>>>>>>>>>>>>>>>>>>>>>>>>>>>>>>>>>>>>>>>>>>>>>>>>>>>>>>>>>>>>>>>>>>>>>>>>>>>>>>>>>>>>>>>>>>>>>>>>>>>>>>>>>>>>>>>>>>>>>>>>>>>>>>>>>>>>>

Reviewer #2: Thank you for the submission of a well-structured and comprehensive investigation into the prevalence of Bartonella spp. DNA in dogs with splenic hemangiosarcoma, highlighting important methodological considerations regarding sample preservation and PCR detection efficiency. The findings significantly contribute to the understanding of pathogen detection in archived tissue samples and offer valuable guidance for future diagnostic studies. The manuscript is well-written, and the data analysis is robust. I appreciate the thorough approach and clear presentation of results. I recommend acceptance with no further revisions.

6. PLOS authors have the option to publish the peer review history of their article (what does this mean? ). If published, this will include your full peer review and any attached files.

**Do you want your identity to be public for this peer review?** For information about this choice, including consent withdrawal, please see our Privacy Policy .

Reviewer #1: **Yes: ** Qaes Talb Al-Obaidi

Reviewer #2: No

---

## [Editor Report · Acceptance letter]

PONE-D-25-00356

PLOS ONE

Dear Dr. Breitschwerdt,

I'm pleased to inform you that your manuscript has been deemed suitable for publication in PLOS ONE. Congratulations! Your manuscript is now being handed over to our production team.

Kind regards,

on behalf of

Dr. Dina Aboelsoued

Academic Editor

PLOS ONE